# Microbiological Evaluation of the Disinfecting Potential of UV-C and UV-C Plus Ozone Generating Robots

**DOI:** 10.3390/microorganisms9010172

**Published:** 2021-01-15

**Authors:** Angel Emilio Martínez de Alba, María Belén Rubio, María Eugenia Morán-Diez, Carlos Bernabéu, Rosa Hermosa, Enrique Monte

**Affiliations:** 1Spanish-Portuguese Institute for Agricultural Research (CIALE), Department of Microbiology and Genetics, University of Salamanca, Campus de Villamayor, 37185 Salamanca, Spain; belenru@usal.es (M.B.R.); me.morandiez@usal.es (M.E.M.-D.); rhp@usal.es (R.H.); emv@usal.es (E.M.); 2Arborea Intellbird, Science Park University of Salamanca, Campus de Villamayor, 37185 Salamanca, Spain; carlos@aracnocoptero.com

**Keywords:** UV-C, ozone, robotics, surrogate microorganisms, disinfection, *Trichoderma*

## Abstract

This study examined the microbicidal activity of ultraviolet (UV)-C_185–256-nm_ irradiance (robot 1) and ozone generated at UV-C_185-nm_ by low-pressure mercury vapor lamps (robot 2) adapted to mobile robotic devices for surface decontamination, which was achieved in less than 1 h. Depending on their wall structure and outer envelopes, many microorganisms display different levels of resistance to decontaminating agents. Thus, the need for novel disinfection approaches is further exacerbated by the increased prevalence of multidrug-resistant bacteria, as well as the potential of novel microorganisms, with the ability to cause disease outbreaks. To set up a rapid and effective approach for microorganisms propagation prevention, we focused on the effects of UV-C and ozone on a distinct microorganism survival ratio. A set of microorganisms, including *Escherichia coli*, *Micrococcus luteus*, *Saccharomyces cerevisiae*, *Trichoderma harzianum*, and *Bacillus subtilis,* were used to evaluate the disinfection power of UV-C and UV-C plus ozone generating robots. UV-C disinfection can be suited to ad hoc tasks, is easy to operate, requires low maintenance, does not have the need for the storage of dangerous chemicals, and does not produce by-products that may affect human health and the environment. The robotic cumulative irradiation technology developed (fluence accumulated values of 2.28 and 3.62 mJ cm^−2^, for robot 1 and 2, respectively), together with the production of ozone (with a maximum peak of 0.43 ppm) capable of reaching UV-C shaded surfaces, and analyzed in the current study, despite being designed for the need to reduce the risk of epidemic outbreaks in real-life scenarios, represents a versatile tool that could be employed for air and surface disinfection within many circumstances that are faced daily.

## 1. Introduction

Microbiological disinfection hierarchy is a concept based on the general descending order of susceptibility of classes of microorganisms to antimicrobial chemicals. Disinfection kills most, but not necessarily all, microorganisms, and depending on the efficacy of the methods and the resistance of the microorganisms, two types of disinfection hierarchy are usually established, namely: (i) the one that evaluates disinfection systems and their efficacy, starting from cleaning and disinfectant cleaning (hygienization) to sanitization and disinfection, until reaching sterilization, which kills all microorganisms, and (ii) the one that is distinguished by the placement across levels different microorganisms depending on their different structures, compositions, and physiologies which determines the order of resistance to antiseptics and disinfectants [1]. Most microbial-based hierarchical distributions are built according to the variety of susceptibilities of human pathogens to chemical disinfectants. However, there seems to be a constant pattern in which the highest resistance corresponds to prions, protozoa cysts (i.e., coccidia), and bacterial endospores (i.e., *Bacillus*), while the lowest resistance corresponds to lipid enveloped viruses (coronavirus, retrovirus, orthomyxovirus, etc.) that always have a lower degree of resistance than Gram-negative bacteria [1,2]. For biosafety reasons, the hierarchical scales are not normally composed of potentially dangerous pathogens (such as the Ebola virus or SARS-CoV-2) but use surrogate microorganisms. Approaches with surrogate microorganisms are based on the axiom that the experimental disinfection or sterilization action on a hardier microorganism to physical treatments and chemical agents, including heat, drying, radiation, and chemicals, will necessarily be more effective against less resistant ones.

Ultraviolet (UV) light has been demonstrated to be capable of destroying viruses, bacteria, and fungi [3,4]. The SARS-CoV-2 virus has not yet been specifically tested for its UV susceptibility, but other tests on related coronaviruses have concluded that they are highly susceptible to UV inactivation [5,6]. The use of UV-C (100–280 nm) irradiation has an advantage with respect to higher (280–400 nm) wavelength UV treatments (A and B) because of its higher DNA damaging energy, which results in the formation of photodimers in the genomes of microorganisms [4,7]. UV-C has been used for no-touch surface decontamination of multidrug-resistant bacteria in hospital settings [8,9,10]. UV-C has also been incorporated in continuous air disinfection systems, allowing for effective air purification by decreasing viable airborne microorganisms [11]. In vitro investigations have reported that extremely small doses of 1.2 to 1.7 mJ cm^−2^ of far UV-C_222-nm_ light inactivated aerosolized H1N1 influenza virus and coronaviruses [6,12]. It has been reported that UV-C_207–222-nm_ is a promising candidate against COVID-19 propagation, as this radiation would inactivate SARS-CoV-2 and other hazardous microorganisms that contaminate surfaces in public spaces without harming mammalian skin or eyes [13]. Ozone produced from ambient air can be considered a side effect of far UV-C lamp operation, although it is well known for its killing action against bacteria, fungi, and viruses [14,15]. Furthermore, the combined application of UV-C and ozone has demonstrated synergistic effects against *Escherichia coli* [16], *Bacillus subtilis* endospores [17], and *Aspergillus niger* conidia [18].

In the present work, the disinfecting power of low-pressure mercury vapour lamps (displayed on two Arborea Intellbird robots, namely: robot 1 with filter glass 40 W UV-C_245–256-nm_ light, and robot 2 with quartz glass 450 W UV-C_185–254-nm_ light and ozone generation at a 185 nm wavelength) was evaluated on four artificially contaminated surfaces (wall, window, floor, and table). The evaluation was performed to assess the disinfection power of the robots on microorganisms with different degrees of resistance. Our objective was to obtain high-level disinfection (HLD) through the complete elimination of all viable microorganisms, except for some fungal conidia and bacterial endospores, when they were present in a significant amount, on the four inanimate surfaces considered after 1 h of dynamic exposure to UV-C and UV-C plus ozone generated by robots 1 and 2, respectively.

## 2. Materials and Methods

### 2.1. Robot 1

The first robot was based on utility models registered by Arborea Intellbird (Salamanca, Spain) and was built by modifying the “Conga” household cleaning robot with the authorization and collaboration of its manufacturer (Cecotec, Valencia, Spain). The robot is disc-shaped at the base, measuring 35 cm in diameter and 110 cm in height (Figure 1). This model of cleaning robot was originally designed for vacuuming and washing floors. It is a light platform, with remarkable robustness, with “detect and avoid” and “detect and actuate” sensors that allow the robots to move with complete freedom. The system includes a remote sensing tool that uses Laser Imaging Detection and Ranging (LIDAR) to illuminate the target with a laser light and measures the reflection (0.3 m s^−1^) to build a map of the environment, facilitating the precise positioning of the robot. The HLD system uses a filter glass 40 W UV-C_245–256-nm_ mercury amalgam luminaire. Lamps are anchored to the base through two UV-resistant plastic adaptors and stainless-steel rods. The equipment works automatically, crossing the surface of the environment to be disinfected without human presence with great precision. It automatically charges the battery when it runs out and resumes work at the point where it stopped. Disinfection is achieved through the accumulation of UV-C radiation peaks, with at least five runs at less than 1 m (20, 38, 56, 74, and 92 cm) distance from the target. 

### 2.2. Robot 2

Based on the same robotic platform as robot 1, but measuring 148 cm in height (Figure 2), this second robot has a power of 450 W in the germicidal luminaires with quartz glass UV-C_185–254-nm_ light. In addition to generating UV-C in the germicidal band close to 253 nm, it also produces light in the ozone spectra, generating a 185 nm wavelength. This has a double germicidal effect on the surfaces that need to be disinfected. The ozone peak measured on the target surfaces, at the time of passing this robot at a 20 cm distance, is 0.37 part per million (ppm).

### 2.3. Microorganisms and Culture Conditions

In this study, strains from the Department of Microbiology and Genetics teaching collection (University of Salamanca, Salamanca, Spain), namely, *Escherichia coli* MG01 (Gram-negative bacterium), *Micrococcus luteus* MG04 (Gram-positive coccus), *Bacillus subtilis* MG07 (endospore-forming Gram-positive bacillus), and *Saccharomyces cerevisiae* MG10 (yeast), and the Spanish Type Culture Collection (CECT, Valencia, Spain) strain *Trichoderma harzianum* CECT2413 (sporulated filamentous fungus), were chosen as model microorganisms representing distinct ranges of survival against disinfectants, thus being suitable indicators and an adequate measure of overall microorganism survival determination. A pure culture of each microorganism was incubated either on Luria–Bertani (LB) broth plates containing 1.5% agar or on Potato Dextrose Agar (PDA) plates, at 37 and 28 °C, respectively. After five days of growth, the microorganisms were collected by scrapping with an inoculation loop, resuspended in sterile water, and diluted to a 0.5 OD_600-nm_ (Zuzi spectrophotometer, Rogo-Sampaic, Wissous, France). Then, the number of viable bacteria or fungi in the microbial suspensions was estimated using the method of counting colony forming units (cfu) and, expressed as cfu mL^−1^, corresponded to the following: *E. coli* 1.65 × 10^9^, *M. luteus* 3.1 × 10^6^, *S. cerevisiae* 1.6 × 10^4^, *T. harzianum* 1.25 × 10^7^, and *B. subtilis* 1.93 × 10^8^. These suspensions were the standardized microbial suspensions that were kept at 4 °C and were used in further experiments.

### 2.4. Sampling on Inoculated Surfaces 

A prospective study was conducted to evaluate the effectiveness of a UV-C light no-touch disinfection robot at reducing environmental contamination. Before UV-C irradiation, 1 mL or 0.1 mL of microbial suspension were placed in an area of 9 cm in diameter (63.6 cm^2^) for irradiation. Sampling was performed on four different surfaces: two horizontal surfaces, namely a melamine table (T) and vinyl floor (F), and two vertical surfaces, namely a plastic painted wall (W) and glass window (G). Floor samples were taken 10 cm away from the wall. Wall and glass window samples were 110 cm and 85 cm, respectively, from the ground. The height of the table was 73 cm. In T, F, and W surfaces, a 1 mL from each microbial suspension was applied using a sterile cotton swab. A similar procedure was followed for the G surface but applying only 0.1 mL from each microbial suspension because of the difficulty in holding such a volume on a vertical glass surface. All of the surface inoculations were carried out in triplicate. 

### 2.5. UV-C Irradiations

All of the UV-C irradiations were conducted in the absence of people, inside a 135 m^3^ (270 cm in height, 700 cm in width, and 714.5 cm in length) room with no controlled climate; thus, disinfection was carried out for 1 h at room temperature, between 18 and 21 °C. The UV-C irradiance of robot 1 (245–256 nm) and robot 2 (185–254 nm) was measured as µW cm^−2^ using a Linshantech UV Light Meter LS126C (Shenzen Linshang Technology, Huizhou, China), which measures 254 nm UV-C light in a range of 0–20,000 μW cm^−2^, with a spectral response from 230 to 280 nm and a dominant wavelength of λp = 254 nm. The irradiance value measurements were converted into units of fluence (mJ cm^−2^). The LS126C apparatus was located on the target surface while the robots operated freely around the room. The UV-C sensor measured the light perceived at each pass of the robot at less than 1 m distance. This resulted in five measurements because each measure was taken 18 cm further away from the previous one, which were 20, 38, 56, 74, and 92 cm from the surface on which the sensor was placed. The robots operated for 1 h inside the 135 m^3^ room where the study was carried out. As one of the walls of the room was made of glass, sensor-recorded measurements could be taken at each pass in front of the robot throughout the experiment. The complete operation of robot 1 can be seen in Appendix A

### 2.6. Ozone Treatments

As for the UV-C irradiations, all of the ozone treatments were conducted in the above-described room and in the absence of people for obvious safety reasons. A cardboard sheet with an aluminum layer on its surface was placed 0.5 cm away from plastic painted wall surfaces where an *E. coli* suspension had been applied. In such a manner, the wall surfaces covered by the cardboard could not be irradiated by the UV-C light generated by robot 2, yet the ozone produced at 185 nm could reach them. Ozone production was monitored for 2 h with a portable WatchGas UNI O3 (WatchGas, Capelle aan den Ijssel, The Netherlands), with a resolution of 10 ppb (0.01 ppm) ozone. An ozone sensor was placed on the wall surface where the reference microorganisms were distributed and were covered with aluminum foil in order to avoid exposure to UV-C light. Although the robot was working for 50 min (when it was switched off), the ozone-meter was working for 120 min, until we found out that the ozone levels were undetectable by the device. As indicated for the UV-C sensor, one of the walls of the room was made of glass, which allowed us to see the sensor-recorded measurements taken throughout the experiment. Thus, ambient ozone concentrations were monitored during the whole procedure. The levels of ozone in the room were 0 ppm as the ozone-meter did not register the level of the gas just before the robot began to work. A maximum peak was obtained at 0.43 ppm, as can be seen in Figure 3.

### 2.7. Microbial Sampling

Un-irradiated (N0) and irradiated samples (N) were collected separately before (before irradiation) and after exposure to UV-C light. For enumeration, both N0 and N (immediately after irradiation and after exposure to ozone, when applicable) samples from each model microorganism and type of surface were collected on the replicate organism detection and counting (Rodac) plates in triplicate. Five sets of six 9 cm diameter circles, three for N0 and three for N samples, were marked on the surfaces to be tested (Appendix A). The circles served to determine the application area of the microbial suspensions and the exact points where the 6.6 cm diameter Rodac plates had to be pressed to the surface for 5 s for the microorganism’s recovery. Additionally, samples from the four analyzed surfaces where the model microorganisms had not been applied were taken in triplicate to evaluate the UV-C effect on environmentally present microorganisms. Rodac-plate count agar (PCA) plates were used to count bacteria, and Rodac-Rose Bengal chloramphenicol (RBC) plates were chosen as the selective medium for the enumeration of filamentous fungi and yeast. Bacterial plates were incubated at 37 °C for 24 h (robot 1) and 48 h (robot 2). The fungal plates from both assays were incubated at 28 °C for 48 h. All of the plates were considered for analysis, and no further clear bacterial or fungal colonies were observed after this incubation time.

### 2.8. Disinfection Experiments Analysis

The disinfection power of both robots was evaluated, considering the cfu reduction levels of the target microorganisms. The changes in cfu after UV-C or UV-C plus ozone treatments were quantified by determining the number of survival microorganisms. This rate expresses the performance as a percentage reduction in terms of a reduction factor, and, for convenience, typically in factors of 10 using a logarithmic (log) reduction scale—a log reduction value (LRV). Log reduction is a mathematical term that is used when testing a product’s efficacy in order to show the relative number of live microorganisms eliminated from a surface through disinfection. This value can be expressed as a LRV function (Log reduction = log_10_ (N_0_/N)) with respect to the initial microorganism concentration before the disinfection process. Where N_0_ is the cfu of the microorganisms before UV-C irradiation or UV-C irradiation plus ozone exposure (cfu mL^−1^), and N is the cfu of the microorganisms after UV-C irradiation or UV-C irradiation plus ozone exposure (cfu mL^−1^). For example, a 1 log reduction corresponds to inactivating 90% of a target microorganism, with the microbial count being reduced by a factor of 10. Thus, a 2 log reduction will see a 99% reduction, or microorganism reduction by a factor of 100, and so on. Bacterial or fungal counts over 100 cfu per 25 cm^2^ (which is the area of the Rodac plate) were considered uncountable, and such a value, according to ISO 100.012, has to be below 10 cfu per 25 cm^2^ once the disinfection process has concluded.

## 3. Results

### 3.1. Robot Operation 

The agility and small size of the equipment allowed the filter glass UV-C_245–256-nm_ and quartz glass UV-C_185–254-nm_ low-pressure mercury vapor luminaires to be positioned very close to the surfaces to be disinfected, working from different angles because of the ease of the automatic advance of the robots, which allowed for the accumulation of a minimum of five UV-C irradiation peaks on the same target point at a distance of 1 m or less, with 1.21 and 1.70 mJ cm^−2^ at 20 cm for robots 1 and 2, respectively. Table 1 shows the cumulative UV-C doses detected on a target point of the wall surface after five runs within a 1-m distance at 20, 38, 56, 74, and 92 cm from robots 1 and 2, with fluence accumulated values of 2.28 and 3.62 mJ cm^−2^ for robots 1 and 2, respectively. The use of mobile devices served to avoid the problems of static UV-C equipment, which, despite their high emission power, achieved lower radiation intensities on the surfaces and left the shaded areas without disinfection. The UV-C fluence values shown in Table 1 were obtained in the 135 m^3^ room in which the robots were working for 1 h, with indications of different distances between the luminaire and sensor.

### 3.2. Robot 1 Microbial Validation

The experimental design followed validated the surface-disinfecting power of the 40 W UV-C robot 1, and the results obtained are shown in Figure 1. After 24 h of incubation on Rodac-PCA and 48 h of incubation on Rodac-RBC plates, respectively, the log reduction means in the cfu bacterial counts on the Rodac-PCA, and the fungal counts on the Rodac-RBC plates, for each one of the three replicates, with an indication of the standard deviation and population reduction percentage of each microorganism tested on four surfaces, were obtained (Appendix A). When cfu overlaps were observed, preventing reliable counting, the data were annotated as “countless” and were assigned a value of log_10_ 3 for analytical purposes. Finally, to calculate the log_10_ reduction, microbial values ranging from 1.6 × 10^4^ cfu mL^−1^ (*S. cerevisiae*) to 1.65 × 10^9^ cfu mL^−1^ (*E. coli*), as well as median values obtained after the irradiation process, were used. The determined log_10_ reduction data are depicted in Table 2. Some values were explained by the initial concentration of inoculum, with higher log_10_ reduction values being reached. Obviously, *S. cerevisiae* and *M. luteus*, with a starting concentration of 1.6 × 10^4^ cfu mL^−1^ and 3.1 × 10^6^ cfu mL^−1^, cannot have a log_10_ reduction above 4 or 6, respectively. Likewise, both in the environmental pollution controls and the bacterial and fungal artificially-inoculated surfaces, the recorded cfu reduction numbers validate the efficacy of robot 1 to achieve HLD of vertical and horizontal surfaces made of different materials.

### 3.3. Robot 2 Microbial Validation

Similarly, the experimental procedure to validate the surface-disinfecting power of the 450 W UV-C robot 2, and the results obtained are shown in Figure 2. After 48 h of incubation on Rodac-PCA and Rodac-RBC plates, the separate cfu bacterial and fungal counts were recorded in triplicate with an indication of the standard deviation and population reduction percentage of each microorganism tested on four surfaces were obtained (Appendix A). Counting criteria were as indicated above for robot 1. The log_10_ reduction values obtained with robot 2 were higher than those obtained with robot 1, describing a higher disinfection power. The obtained log_10_ reduction data are depicted in Table 2, and, as for robot 1, satisfactory microbial disinfection was achieved. As observed for robot 1, the log_10_ reductions were the maximum that could be reached from the initial concentration values, indicating reduced microbial survival.

With the aim of determining the ozone contribution to the surface-disinfecting power of robot 2, similar experiments as those described above were carried out. However, unlike previously done, the surfaces were covered for the current assays, thus avoiding UV-C radiance interference. The ozone produced by the 450 W UV-C robot 2 was monitored for 2 h (Figure 3), obtaining a maximum peak of 0.43 ppm that coincided with the moment at which the robot turned off (50 min). However, values above 0.1 ppm were registered for another 70 min. The US Occupational Safety and Health Administration (US-OSHA) Threshold Limit Value-Time Weighted Average (TLV-TWA) is 0.1 ppm, and it is defined as the ozone concentration to which individuals can be repeatedly exposed for a normal 8 h workday [19]. Nonetheless, the *E. coli* values employed for measuring the UV-C plus ozone effect (1.65 × 10^9^ cfu mL^−1^) were too high for observing an ozone effect. A lower value was employed (1.65 × 10^7^ cfu mL^−1^) to determine the ozone surface-disinfecting power. The results obtained showed a log_10_ reduction of 5.31. After contrasting the recorded values with the resistance that could be expected for the surrogate microorganisms, the disinfection power of the combined application of the 450 W UV-C irradiation plus ozone could be considered satisfactory for the surfaces tested.

## 4. Discussion

A collaborative investigation has been carried out harnessing robotics, engineering, and microbiological efforts to produce and validate automated and self-sufficient systems able to generate HLD power in our daily life for ordinary touched surfaces during the COVID-19 lockdown. Both in the environmental pollution controls and the bacterial and fungal artificially inoculated surfaces, microbial population reduction values have served to validate the effectiveness of the robotic solutions through applying 40 W UV-C light and 450 W UV-C light, the latter with the capacity to generate 0.43 ppm ozone peaks, so as to obtain HLD on vertical and horizontal surfaces made of different materials. The result has been that the effectiveness of the two developed robots was proven, validated by their effect in reducing the populations of surrogate microorganisms with different degrees of resistance to UV-C radiation and ozone. The hierarchical resistance of microorganisms varies according to the type of antiseptic and disinfectant tested. Some authors have found the resistance of Gram-negative bacteria, with a thin but complex cell wall, to be above yeasts, as well as Gram-positive bacteria, to be ranked from least to most susceptible to chemical disinfection [1]. However, there is no question that enveloped viruses have the lowest scores in disinfectant resistance tests [2]. The US environmental protection agency (EPA) has also used the disinfection hierarchy to approve the use of registered products for treating surfaces contaminated with emerging viral pathogens, specifically on the SARS coronavirus (SARS-CoV), influenza A (H1N1) virus, and Ebola virus [20]. The reduction percentages obtained with subrogate microorganisms with much more recognized resistance than lipid-enveloped viruses, such as SARS-CoV-2, reaffirms the effectiveness achieved in real-life scenarios of our disinfection systems with accumulated UV-C plus ozone. It should be pointed out that it was not necessary to apply irradiation cycles and doses to a clean room and operating room control, but instead the application of a cycle of 1 h with enough UV-C plus ozone to control microorganisms (viruses, bacteria, fungi, and spores), for use in daily life.

The outcome of the current study, although aimed at the need to face the new threat of SARS-CoV-2, represents a versatile tool for broader use. Thus, the obtained data validate the employment of UV-C and UV-C plus ozone generating robots described in the current study for the disinfection of different surfaces, spaces, and environments. In this sense, it is noteworthy to mention that the results obtained in one of the surfaces (wall) are comparable to those achieved for glass. Indeed, this result is relevant because of the different nature of both surfaces, one being more porous (wall) than the other (glass). It would be reasonable to think that in porous surfaces, microorganisms would last longer as the penetrance of the UV-C light is not so good, and the cavities present in such surfaces could provide a reservoir. This is something that could be faced by the UV-C plus ozone generating robot and presumably would penalize the disinfection power of UV-C robot lacking ozone generation. However, our data refute such a theory, as the results achieved with both robots showed similar results in the analyzed surfaces. Every microorganism, based on its biological characteristics, has a distinctive sensitivity in response to a UV-C dose, a kind of fingerprint. Whereas *B. subtilis* requires more energy to achieve inactivation, microorganisms like *E. coli* and enveloped viruses become inactivated with a relatively low dose of radiance. However, different works have reported that different experimental conditions make it difficult to achieve a general UV-C user criterion. By applying higher doses of energy, reducing exposure distance, and increasing exposure times, the disinfection ratio is greatly improved. In addition, the dose of static UV-C is determined based on the intensity of a given UV-C wavelength at a fixed distance. In this way, UV-C_253.7–254-nm_ at fluence rates of 40.4 to 50 mJ cm^−2^ inactivated endospores of *B. subtilis* [21,22], and 15 s of UV-C_222-nm_ at 29.2 mJ cm^−2^ was able to achieve a 100% control of *E. coli* [23]. Similar UV-C_254-nm_ doses were needed for 90% inactivation of *B. subtilis* (48 mJ cm^−2^) and *E. coli* (30 mJ cm^−2^) [24]. By comparison, much lower fluence rates of 1.1 mJ cm^−2^ of UV-C_254-nm_ [25] and 1.6 mJ cm^−2^ of UV-C_222-nm_ [12] served to inactivate 95% aerosolized H1N1 influenza virus. This last study concluded that low-intensity levels equivalent to 2 mJ cm^−2^ of UV-C_222-nm_ on the center of the irradiation target in public locations might represent a safe and efficient methodology for limiting the transmission and spread of airborne-mediated microbial diseases. This same UV-C_222-nm_ fluence was demonstrated to produce enough DNA damage to kill *B. subtilis* endospores [26]. Although this particular study also reported skin damage after typical UV-C_254-nm_, none after UV-C_207-nm_ [26], our robots were designed to be used exclusively in the absence of people, and the sweeping of the entire room by the robot achieved a cumulative effect of additional radiation in successive passes. Exposure time plays an extremely important role, and the dynamic UV-C_245–256-nm_ and UV-C_185–254-nm_ lamps were able to build up enough fluence, 1.21 mJ cm^−2^ for robot 1 and 1.70 mJ cm^−2^ for robot 2 in just one run at a 20 cm distance, to attain surface HLD without any doubt whatsoever. A recent report has shown that the combination of UV-C light-emitting diodes (LEDs), or low-pressure mercury vapor UV-C_254-nm_ lamps with 2 mg L^−1^ chlorine, effectively inactivated waterborne spores of *T. harzianum* [27]. In the present study, as shown in Table 2, we have proven that both robots, with a successful disinfection effect, displayed a better microbial control performance on vertical surfaces. As we have started out from different microbial densities, 10^9^ cfu mL^−1^ for *E. coli* to 10^4^ cfu mL^−1^ for *S. cerevisiae*, the log_10_ reduction values calculated for both robots were proportionate to the initial microbial quantities tested. Thus, the fact that *S. cerevisiae* was reduced by 1.2 log_10_ after robot 1 action on horizontal surfaces does not indicate more resistance of this yeast to UV-C_245–256-nm_ than *B. subtilis* endospores, with reductions of 5.28–5.80 log_10_, as the reduction range was much lower for *S. cerevisiae*, starting from a much smaller amount. Similarly, the log_10_ reduction values obtained for *M. luteus* can be explained by its starting density, which was from 62 to 532 times lower than those of *B. subtilis* and *E. coli*, respectively.

Ozone is a toxic gas that is effective as a germicide, but it must be present in a concentration far greater than that which can be safely tolerated by humans or animals. The inhalation of ozone can cause sufficient irritation to the lungs that can result in pulmonary edema [28]. As a result, the maximum acceptable concentration of generated ozone must not exceed 0.05 ppm by volume of air circulating through a given device in the presence of people. In any case, the full killing power of ozone occurs mostly during the first 4 min of exposure, after which it starts to decompose [29]. An early report on non-enveloped poliovirus identified genomic damage as the most likely mechanism of viral inactivation [30]. It is well known that by targeting the membrane glycoproteins, glycolipids, or certain amino acids such as tryptophan, ozone acts on the sulfhydryl groups of certain enzymes and structural proteins, resulting in a disruption of normal microbial activity [31]. Although it has been reported that high ozone concentrations (80–150 ppm) were needed to reduce *B. subtilis* endospores by 6–7 log_10_ under exposure periods of 90 min [32,33], a lower (2.3 ppm) ozone lethal threshold concentration was proposed for *Bacillus* endospores [34]. Furthermore, other authors [35] demonstrated that the *E. coli* minimum bactericidal concentration of ozone is 1 ppm. Although surface disinfection of non-enveloped viruses was obtained with 6.25 ppm ozone when applied alone for 0.5–10 min [36], such a concentration does not contradict that the inactivation could also be achieved with lower doses. Exposure of contaminated surfaces to 2 ppm ozone for 4 h has been proposed as an extensive and effective supplement to traditional chemical disinfection methods [37]. The synergistic bactericidal effect of combined UV-C and ozone owing mainly to the enhanced destruction of *B. subtilis, E. coli*, or *A. niger* cell structures by producing excess hydroxyl radicals is well documented [16,17,18]. In our study, the UV-C_185-nm_ ozone generation (0.22 ppm in a 1-h cycle with a maximum peak of 0.43 ppm) contributed to the disinfection power of the cumulative 3.62 mJ cm^−2^ UV-C_185–254-nm_ fluence produced at <92 cm distance from the target surface. The ozone produced had, by itself, a disinfection power generating a log_10_ reduction of 5.31. That implies that areas that could not be reached by the UV-C light could nevertheless be disinfected. Higher range ozone-producing robots will help to eliminate microorganism reservoirs, despite staying in the shadows. The US-OSHA requires that workers not be exposed to an average concentration of more than 0.10 ppm for 8 h. As ozone is highly reactive, the 0.22 ppm produced by robot 2 rapidly dissipated in a 135 m^3^ room, reaching levels below the US-OSHA limit, namely: 0.1 ppm after 120 min, and 0 ppm at 140 min, making it safe for everyday use, after a 1 h safety period. 

Here, we have used a set of microorganisms that can be hierarchically classified according to their level of resistance to evaluate the efficacy of two robots equipped with UV-C lamps and ozone production in order to obtain HLD on four inanimate surfaces after 1 h of dynamic exposure. This technology has been independently successfully used and has proven to be effective when used in nursing homes during the COVID-19 outbreak, and the robotic solution represents a versatile tool that could be employed for disinfection in the light of microbial threats posed by our everyday life. Because of their small size and easy handling, the two robots have great potential to be used in all closed environments where the transmission of pathogens is maximum, and, despite the fact that they cannot be employed in the presence of people, the short time needed to achieve HLD outweighs such a constraint. Disinfection is one of the cornerstones of infection prevention and, therefore, disease control. Thus, apart from maintaining areas with a low level of microorganisms routinely when needed, like surgical rooms, UV-C generating ozone robots will be of great help to prevent pathogens spreading and will be a useful tool during outbreaks and pandemics, becoming a first line of defence against pathogens propagation.

## Figures and Tables

**Figure 1 microorganisms-09-00172-f001:**
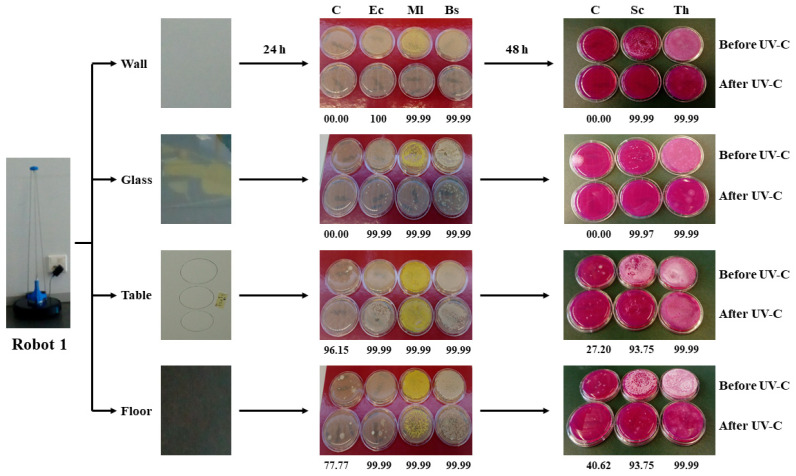
Scheme of representative surfaces showing the robot 1 apparatus and area of sampling. Percentage of reduction survival values based on bacterial and fungal colony forming unit (cfu) counts after 24 h incubation of Rodac-plate count agar (PCA), and 48 h incubation of Rodac-Rose Bengal chloramphenicol (RBC) plates, respectively, are depicted. To calculate the log_10_ reduction, starting microbial values ranging from 1.6 × 10^4^ cfu mL^−1^ (*Saccharomyces cerevisiae*) to 1.65 × 10^9^ cfu mL^−1^ (*Escherchia coli*), as well as median values obtained after irradiation processes were used. As a log_10_ value of 3 was given to plates when the cfu values overlapped, a high percentage of reduction could be determined where no reduction in counts seemed to be visually appreciated when comparing before and after plates, that is due to the impossibility of cfu counting and the arbitrary number given to those plates for analytical purposes. Acronyms refer to Control (C), *E. coli* (Ec), *Micrococcus luteus* (Ml), *Bacillus subtilis* (Bs), *S. cerevisiae* (Sc), and *Trichoderma harzianum* (Th).

**Figure 2 microorganisms-09-00172-f002:**
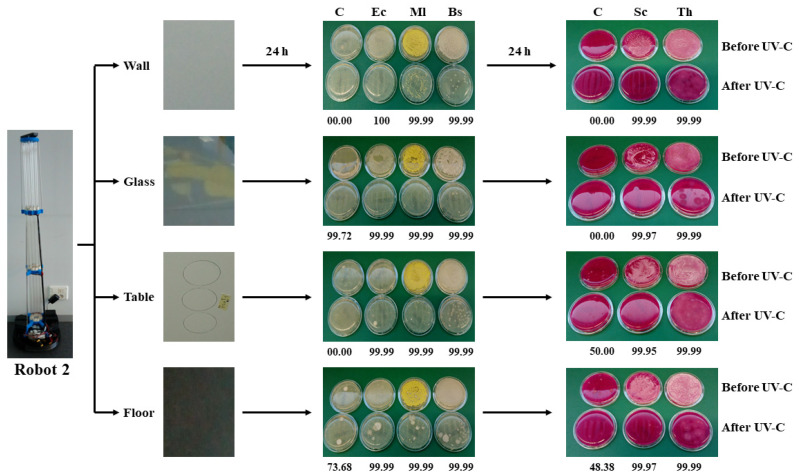
Scheme of representative surfaces showing the robot 2 apparatus and area of sampling. Percentage of reduction survival values based on bacterial and fungal cfu counts after 48 h incubation of Rodac-PCA, and Rodac-RBC plates, respectively, are depicted. As for robot 1, to calculate the log_10_ reduction, starting microbial values ranging from 1.6 × 10^4^ cfu mL^−1^ (*S. cerevisiae*) to 1.65 × 10^9^ cfu mL^−1^ (*E. coli*), as well as median values obtained after irradiation processes were used. As a log_10_ value of 3 was given to plates when cfu overlapped, a high percentage of reduction could be determined where no reduction in counts seemed to be visually appreciated when comparing before and after plates, that is due to the impossibility of cfu counting and the arbitrary number given to those plates for analytical purposes. Acronyms refer to Control (C), *E. coli* (Ec), *M. luteus* (Ml), *B. subtilis* (Bs), *S. cerevisiae* (Sc) and, *T. harzianum* (Th).

**Figure 3 microorganisms-09-00172-f003:**
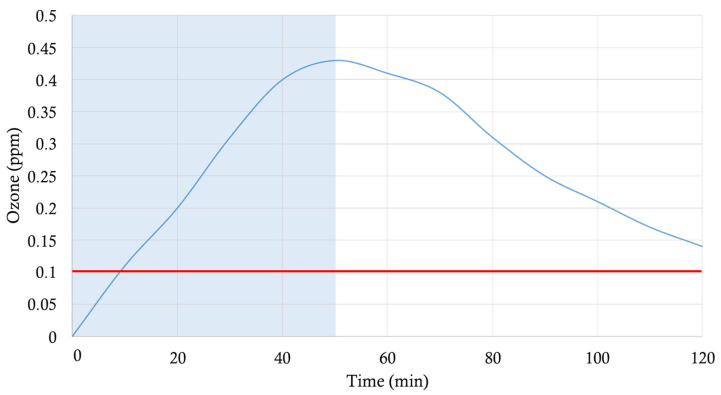
Ozone generated (ppm) by robot 2 (450 W, UV-C_185–254-nm_) with indications for functioning starting time (0 min), and robot switching off (50 min), depicted by a shadowed area, and measurement ending (120 min). In red is the US- Occupational Safety and Health Administration (OSHA) Threshold Limit Value (0.1 ppm).

**Table 1 microorganisms-09-00172-t001:** UV-C fluence (mJ cm^−2^) generated by robot 1 (40 W, UV-C_245–256-nm_) and robot 2 (450 W, UV-C_185–254-nm_) with an indication of the distance (20, 38, 56, 74, 92, 100, and 200 cm) between luminaire and sensor. The accumulated value corresponds to the sum of five measurements obtained from the five runs of the robot being each radiation measured 18 cm further away from the previous one (20, 38, 56, 74, and 92 cm).

Distance (cm)	20	38	50	56	74	92	Accumulated	100	200
Robot 1 Fluence (mJ cm^−2^)	1.21	0.50	0.38	0.27	0.17	0.12	2.28	0.12	0.03
Robot 2 Fluence (mJ cm^−2^)	1.70	0.81	0.62	0.50	0.35	0.26	3.62	0.26	0.09

**Table 2 microorganisms-09-00172-t002:** Log_10_ reduction values in microbial viability after robot 1 (40 W, UV-C_245–256-nm_) and robot 2 (450 W, UV-C_185–254-nm_) performance. Log_10_ reductions are given for the 1 mL of microbial suspension applied onto all surfaces, apart from Glass (*) where only 0.1 mL from each microbial suspension was evaluated.

	Wall		Glass *		Table		Floor	
	Robot 1	Robot 2	Robot 1	Robot 2	Robot 1	Robot 2	Robot 1	Robot 2
Control (PCA)	0.00	0.00	0.00	2.08	1.41	0.00	0.65	0.73
*E. coli* (1.65 × 10^9^)	9.28	9.22	7.33	7.43	6.22	8.08	8.23	8.23
*M. luteus* (3.10 × 10^6^)	6.49	6.49	5.37	4.63	4.28	6.37	6.12	5.57
*B. subtilis* (1.93 × 10^8^)	6.17	7.35	5.37	6.56	5.28	6.57	5.80	6.61
Control (RBC)	0.00	0.00	0.00	0.00	0.14	0.30	0.23	0.29
*S. cerevisiae* (1.60 × 10^4^)	4.20	4.20	3.20	3.20	1.20	3.36	1.20	3.68
*T. harzianum* (1.25 × 10^7^)	4.10	5.54	3.10	6.10	4.10	4.10	4.10	5.82

## Data Availability

Data is contained within the article or supplementary material.

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
