# Peer review of "Microbiological Evaluation of the Disinfecting Potential of UV-C and UV-C Plus Ozone Generating Robots"

_microorganisms, 2021, doi:10.3390/microorganisms9010172_

Round 1

Reviewer 1 Report

After the revision, the manuscript is better, but only slightly. It is still quite difficult to read due to issue with the English language.  I started making suggested changes, but gave up after the first few pages.

There are significant details in the Methods that still could be better described.  It still is unclear how UV irradiance (and fluence) was measured (where, when).  We know what photometer was used, but that is about it.  The same is true for ozone measurements.  We know what ozone sensor was used, but where were the measurements taken? What was the ambient ozone concentration?  Are you truly able to show that ozone had any meaningful impact on the results obtained?

Still, the biggest issue with the manuscript is that all mention of the Microbial Disinfection Pyramid (MDP) is really irrelevant.  You can develop a MDP to say anything you want, depending on the means of disinfection being tested and the surrogate organisms used.  Sometimes organism groups change places.  The fact that the results confirm a particular MDP is not shocking. Forcing that discussion is a distraction.  Consider eliminating it entirely.  The important part of your work is the inactivation results and the novel way the UV dose was delivered.  Focus on that story and tell it well. Keep in mind that the world already knows that UV can inactivate organisms, that the susceptibility of different organisms is different, and that the level of inactivation achieved depends almost exclusively on the UV dose delivered.  Your work focuses on a novel way to deliver that dose (and a dose in combination with ozone).  Explain further how the robots operated and moved during the experiments.  How long did it take to reach the doses achieved at each distance?  Did the robots go back and forth in a straight line?  What are the implications for disinfecting a real room when the robot decides on its own path? Use the results to support the dose delivery method.  Don’t keep trying to use the results to prove that a particular MPD is correct.  You can make key points about the susceptibility of different microorganisms without all the pyramid language.

If the methods, results, and discussion are tightened up, the work could be worth publishing.

Author Response

New text has been included and modifications have been done following Reviewers advice. See attached document

Reviewer 2 Report

In the manuscript "Microbiological evaluation of the disinfecting potential of UV-C and UV-C plus ozone generating robots" by Martínez de Alba and colleagues, the authors reported on the effect of UV and ozone produced by two robots on different microorganisms, including bacteria, yeast and filamentous fungus. This study is interesting for the readers of Microorganisms. However, the manuscript requires several clarifications.

The abstract is not consistent with the study. The authors mention the coronavirus SARS-CoV-2 when the study does not include this virus, it should focus more on what was done.

Lines152-157: Why are these concentrations of microorganisms used? Why they did not use lower concentrations that would allow better final microorganism counts in all the cases? Is the design of the study based on any standards? Authors should specify this information.

Lines 160-172: In the section "Anaysed surfaces" I understand that the authors also explain the inoculation of the different surfaces when they indicated "microbial suspension were placed in an area...". Therefore, I consider that the title of the section is not the most appropriate and causes confusion.

The results are rather confusing and hard to understand. No cfu values are presented, nor are the standard deviation values considering the three replicates. I consider that these data should be included. Figures 2 and 3 are not clear, moreover, C, Ec, MI...should be specified. It is not clear to me why tables 1 and 2 have two parts. Both parts have similar information, I don't know which part of the table is correct.

In the discussion section there is no strong comparison of the results obtained by the authors with those obtained in other studies. This is an issue that should be improved.

I suggest replacing the word microbe with microorganism.

Author Response

(The authors gave the same response as above.)

Reviewer 3 Report

Martinez de Alba et al. present results of the disinfecting potential of UV-C and UV-C plus ozone generating robots. Methods and results are precisely explained. Evaluation of introduced technique and its efficacy seems realistic. Authors could additionally focus on possible applications during outbreaks and pandemics and also strenghthen limitations given with this technique (penetration depth) within the Discussion section.

Author Response

(The authors gave the same response as above.)
